# Mercury and Autism Spectrum Disorder: Exploring the Link through Comprehensive Review and Meta-Analysis

**DOI:** 10.3390/biomedicines11123344

**Published:** 2023-12-18

**Authors:** Aleksandar Stojsavljević, Novak Lakićević, Slađan Pavlović

**Affiliations:** 1Innovative Centre, Faculty of Chemistry, University of Belgrade, Studentski Trg 12–16, 11000 Belgrade, Serbia; 2Clinical Centre of Montenegro, Clinic for Neurosurgery, Ljubljanska bb, 81000 Podgorica, Montenegro; novak.lakicevic@kccg.me; 3Institute for Biological Research “Siniša Stanković”—National Institute of the Republic of Serbia, University of Belgrade, Bulevar Despota Stefana 142, 11060 Belgrade, Serbia; sladjan@ibiss.bg.ac.rs

**Keywords:** mercury (Hg), autism spectrum disorder (ASD), clinical matrices, comprehensive meta-analysis

## Abstract

Mercury (Hg) is a non-essential trace metal with unique neurochemical properties and harmful effects on the central nervous system. In this study, we present a comprehensive review and meta-analysis of peer-reviewed research encompassing five crucial clinical matrices: hair, whole blood, plasma, red blood cells (RBCs), and urine. We assess the disparities in Hg levels between gender- and age-matched neurotypical children (controls) and children diagnosed with autism spectrum disorder (ASD) (cases). After applying rigorous selection criteria, we incorporated a total of 60 case-control studies into our meta-analysis. These studies comprised 25 investigations of Hg levels in hair (controls/cases: 1134/1361), 15 in whole blood (controls/cases: 1019/1345), 6 in plasma (controls/cases: 224/263), 5 in RBCs (controls/cases: 215/293), and 9 in urine (controls/cases: 399/623). This meta-analysis did not include the data of ASD children who received chelation therapy. Our meta-analysis revealed no statistically significant differences in Hg levels in hair and urine between ASD cases and controls. In whole blood, plasma, and RBCs, Hg levels were significantly higher in ASD cases compared to their neurotypical counterparts. This indicates that ASD children could exhibit reduced detoxification capacity for Hg and impaired mechanisms for Hg excretion from their bodies. This underscores the detrimental role of Hg in ASD and underscores the critical importance of monitoring Hg levels in ASD children, particularly in early childhood. These findings emphasize the pressing need for global initiatives aimed at minimizing Hg exposure, thus highlighting the critical intersection of human–environment interaction and neurodevelopment health.

## 1. Introduction

Autism spectrum disorder (ASD) is a disturbance associated with brain development that causes problems in social interaction and communication, along with restricted and/or repetitive behaviors or interests [1,2]. Terminologically, in 2022, the World Health Organization (WHO) published the latest International Classification of Disease and Related Problems-11th Revision (ICD-11), and the official name of autism is ASD [3]. Additionally, according to the European Autism Information System (EAIS), the official name/observation ASD should be used [4].

The incidence of ASD has risen to such an extent that it has become known as the “ASD epidemic” in scientific literature [5]. According to the latest WHO report, one in every 100 children receives a diagnosis of ASD [6]. In the United States (US), one out of every 36 children carries a confirmed ASD diagnosis [7]. Notably, ASD is considerably more prevalent in boys, with a nearly fourfold higher incidence compared to girls [8,9]. Symptoms of ASD typically manifest within the first year of life, becoming most conspicuous between 18 and 24 months [10,11]. Currently, no prenatal biomarker for ASD exists, and regrettably, ASD remains without a cure. However, with appropriate treatments, it is a condition that can be effectively managed [12,13,14].

It is widely accepted that the etiology of ASD is multifactorial, with various contributing factors [15,16]. In addition to genetic, epigenetic, and certain environmental factors, there is a growing suspicion that non-essential trace metals, particularly those with neurotoxic properties, could play a crucial role in the etiology of ASD. It is known that Hg is not the only toxic metal associated with ASD. Other metals related to ASD are aluminum (Al), antimony (Sb), arsenic (As), beryllium (Be), cadmium (Cd), chromium (Cr), lead (Pb), and nickel (Ni) [17,18]. Non-essential trace elements have no known function in the human body and can be toxic even in low levels. Non-essential trace elements include heavy metals and metalloids such as aluminum Al, As, Cd, Hg, Pb, Sb, tin (Sn), uranium (U), and vanadium (V). Their toxicity is related to their ability to damage vital organs such as the brain, kidney, liver, and others. Long-term exposure to non-essential elements can lead to physical (e.g., chronic pain, changes in blood pressure, changes in blood composition, etc.) and psychological (e.g., anxiety, passivity, etc.) disorders, neurodegenerative diseases, and cancer [19]. Exposure to trace metals commences in utero, as the placental barrier often proves insufficient in preventing their transport to the developing fetus [20]. Furthermore, during the initial year of life, the immature blood–brain barrier leaves infants vulnerable to the effects of non-essential trace metals [21,22]. As a result, trace metals can disrupt critical biochemical processes essential for sustaining life [23]. However, the etiology of ASD is still unclear. Environmental factors that could prenatally influence the onset of ASD include immune abnormalities, zinc deficiency, abnormal melatonin synthesis, maternal diabetes, stress, toxins, and parental age. Postnatal environmental factors include stress, immune abnormalities, and toxic metal. There is extensive evidence of the connection between many prenatal environmental factors and increased development of ASD [24].

The hazardous, non-essential/toxic trace metal Hg has garnered notable attention due to its potent neurotoxic properties and specific neurobiochemistry within the human body. According to its role in the body, Hg is classified as a non-essential toxic element, while it is considered a heavy metal according to its physical and chemical properties [25]. The WHO classifies Hg as one of the 10 priority environmental pollutants [26]. Industrial development has led to a nearly threefold increase in Hg emissions into the environment, with atmospheric Hg levels rising by nearly 1.5% annually [27].

Hg exists in various forms, including elemental (Hg^0^), inorganic (Hg^2+^), and organic (alkyl Hg) forms [28]. While elemental Hg can be ingested orally with minimal adverse effects, it becomes toxic when chemically converted into the Hg^2+^ species [26,29]. Further bioconversion into alkyl Hg results in a highly toxic compound with a strong affinity for lipid-rich organs, specifically the brain [30]. Organic forms of Hg (such as methylmercury, MeHg and ethylmercury, EtHg) are notably more toxic than inorganic forms [31,32].

Even at relatively low levels, Hg can be deleterious, primarily to young children [33,34]. Hg easily crosses both the placental and blood–brain barriers, accumulating in the central nervous system, particularly in the cerebral cortex and cerebellum. MeHg and EtHg are fat-soluble and have a high affinity for thiol groups, allowing them to easily penetrate the placental and the blood–brain barrier [35]. Within cells, Hg binds to mitochondria, the endoplasmic reticulum, and the Golgi apparatus, disrupting their essential biochemical functions [36]. The buildup of Hg in brain structures results in neuroinflammation, oxidative stress, and elevated levels of autoantibodies against brain proteins and other components [37,38]. Chronic childhood Hg poisoning, known as acrodynia or “pink disease”, resulting from exposure to Hg chloride-containing tooth powder, underscores the vulnerability of children to Hg compared to adults [39,40]. Moreover, offspring of acrodynia survivors face a higher risk of developing ASD, with an earlier incidence rate of one in 22 compared to the general child population’s one in 160 [39].

During pregnancy, the main sources of Hg exposure stem from maternal consumption of seafood and the number of dental Hg amalgam fillings [41,42]. Seafood consumption poses a common and potentially hazardous route, primarily due to the ingestion of MeHg through contaminated fish, shellfish, and sea mammals [43]. According to the Food and Drug Administration (FDA), pregnant women, women of childbearing age, and young children are advised to avoid shark, swordfish, mackerel, and tilefish, due to their higher Hg concentrations [44]. In contrast, Hg exposure from dental fillings remains relatively constant over time, with dental Hg amalgam fillings largely being replaced by composite fillings [31]. Airborne Hg exposure, primarily resulting from industrial waste, such as coal burning and mining activities, cannot be ruled out [45,46].

Children have been exposed to Hg later in life through thimerosal, a compound containing EtHg used in some vaccines, and anti-Rho(D) immune globulins for Rh-negative mothers during pregnancy [47,48]. Thimerosal, with approximately 50% of its weight consisting of Hg, has been used as a preservative in numerous vaccines since the 1930s to prevent microbial growth [2,49]. Due to substantial public concern and controversy, the FDA proposed the removal of thimerosal from vaccines between 1999 and 2001 [5]. The CDC, following several multi-year studies, declared that “exposure to thimerosal during pregnancy and in young children was not associated with an increased risk of ASD” [7]. It is worth noting that the incidence of ASD continued to rise even after the removal or reduction of thimerosal from many vaccines in the USA, Europe, and certain Asian countries [7]. Consequently, due to controversy, thimerosal was removed from anti-Rho(D) immune globulins [48]. Presently, the scientific community places substantial focus on the environmental Hg exposure of pregnant women, nursing mothers, and young children, suggesting that environmental triggers could play a more important role in the Hg-ASD link than vaccinations themselves. Kern et al.’s [50] review of 91 papers from 1999 to 2016 found that 74% of these papers identified environmental Hg exposure as a critical risk factor for ASD.

Many animal studies have attempted to determine the neurological mechanisms linking Hg and ASD. Experiments on monkeys show that Hg levels in the brain increase after exposure, and that it is necessary to evaluate the effects of its presence on neurological structures. After administering organic Hg to monkeys, the half-life of Hg in the brain varied considerably in different brain regions. In the thalamus, Hg levels remained the same, and in the pituitary gland, they doubled six months after exposure. Stereologic and autometallographic studies showed that the persistence of Hg in the brain was accompanied by a significant increase in the number of microglia, while the number of astrocytes decreased [51]. An active neuroinflammatory process was detected in the brains of ASD patients, including a marked activation of microglia. Hg-mediated modulation of cytokine production (IL-6, TNF-α) could have an adverse impact on ASD patients, leading to autoimmune brain response, IgG accumulation in brain, and CD4+ T cell infiltration [21]. It is also shown that some cognitive and sensory deficits can be associated with Tryptophan–Kynurenine metabolic system in the human brain [52].

On the other hand, the scientific community’s attention has shifted. The focus is not only on how Hg enters the body, but also on the mechanisms of its removal. This shift is particularly vital in the context of ASD children. Numerous investigations have indicated that this sensitive population group exhibits diminished capabilities in eliminating Hg from their bodies [21,49]. Several important factors have been identified, including heightened levels of oxidized glutathione, the far-reaching consequences of oxidative stress, increased use of oral antibiotics (especially during the first year, which disrupts the gut flora and leads to Hg methylation), alterations in cell cycles, epigenetic modifications (such as histone alterations, DNA methylation, and microRNA expression), as well as antagonistic effects on essential trace elements and changes in the expression of metallothioneins, among others. Further in-depth information on these detrimental effects of Hg on multiple biochemical processes can be found in existing literature [48,53].

The present systematic review and comprehensive meta-analysis aim to examine potential aspects of Hg contributing to ASD in children, investigating Hg levels in different biological materials (hair, whole blood, red blood cells, plasma, and urine), and shed light on the role of Hg levels in the context of this neurodevelopmental condition. To achieve that, we categorize, summarize, and discuss the published research papers on this topic.

## 2. Materials and Methods

The basis for this review and meta-analysis corresponds to the “Preferred Reporting Items for Systematic Reviews and Meta-Analyzes: the PRISMA Statement” [54]. The PRISMA statement was originally proposed in 2009. However, we have utilized the updated PRISMA 2020 statement, which supersedes the 2009 version and incorporates new reporting guidelines that reflect methodological advances in the identification, selection, appraisal, and synthesis of studies.

The main objective of the PRISMA 2020 statement is to ensure that users of review receive a transparent, comprehensive, and accurate account of the rationale for conducting the review, the methodology used, and the findings obtained [55]. It also includes a 27-item checklist, an expanded checklist with detailed reporting recommendations for each item, a PRISMA 2020 abstract checklist, and revised flowcharts for both original and updated reviews.

Prior to commencing the present study, the authors prepared a research protocol.

### 2.1. Information Sources

First, we searched four databases: SCOPUS, PubMed, ScienceDirect, and Google Scholar. However, as the publications in these databases overlap considerably, we concentrated on two of the most representative databases, SCOPUS and PubMed.

### 2.2. Search Strategy

Our major objective in this study was to identify all research that examined Hg levels in hair, whole blood, plasma, RBCs, and urine of neurotypical children (controls) and ASD children (cases). We conducted our literature search from 1985 to the present. To do this, we used a comprehensive search strategy with mesh terms such as “autism”, “autistic”, “child”, “preschool”, “school”, “heavy metals”, “toxic metals”, “mercury”, “Hg”, “hair”, “blood”, “plasma”, “red blood cells”, and “urine”. The authors in our search utilized a total of 1462 ASD-related keywords, with 29 being used most frequently. A graphical representation of these data, illustrating the network visualization and relationships between the observed keywords, can be found in Figure 1A,B (created with VOSviewer 1.6.19 software, copyright (c) 2009–2023 Nees Jan van Eck and Ludo Waltman Centre for Science and Technology Studies of Leiden University, Leiden, The Netherlands).

Additionally, we meticulously reviewed the reference lists of retrieved results. Our inclusion criteria encompassed original, case-control research studies that reported Hg levels in the specified clinical matrices of both cases and controls. Exclusion criteria were studies with adults, studies in which the diagnosis of ASD was not confirmed, studies with cases and controls that were not from the same residence, studies with age- and sex-mismatched cases and controls, studies that reported additional pathologies besides ASD, non-English language studies, studies with insufficient numerical data, and studies with extremely abnormal Hg values. Studies that refer to sufficient numerical data include results where the mean ± standard deviation (SD) or standard error (SE), or some other numerical value from which the SD can be calculated, is accurately reported. In fact, there are many studies that report only the mean without the SD or SE, or studies in which only graphs without numerical values are shown. In our meta-analysis, we set the criterion of using only complete data, i.e., mean values and SDs, that represent sufficient numerical data for us. Further exclusion factors are delineated in Figure 2. For the meta-analysis, we examined original full-length research articles spanning the following timeframes: 1985–2023 for hair (µg/g), 2004–2023 for whole blood (µg/L), 2011–2020 for plasma (µg/L), 2010–2017 for RBCs (µg/L), and 2003–2020 for urine Hg levels (µg/g creatinine). These timeframes were selected to ensure consistency in analytical procedures. Most authors employed inductively coupled plasma mass spectrometry (ICP-MS) to determine Hg concentrations in clinical matrices, while a smaller number utilized atomic absorption spectroscopy (AAS). Two papers used inductively coupled plasma optical plasma spectrometry (ICP-OES), one paper employed the Hg vaporimeter, and one paper utilized atomic fluorescence spectroscopy (AFS).

### 2.3. Study Selection and Data Extraction

A graphical representation of the selection process can be found in Figure 2. Two trained researchers (A.S. and S.P.) independently extracted the following data from each study: author(s) and year of publication, country of origin, sample size (controls/cases), age (controls/cases), gender (number of girls/boys in both groups), type of clinical matrices studied, analytical technique used, and Hg level (mean ± SD, given for controls and cases). In cases where results were reported as mean ± SEM, the SEM (standard error of the mean) was converted to SD using the appropriate formula. Similarly, when authors reported results as an interquartile range (IQR), we converted the IQR to SD [56]. Our inclusion criteria only considered papers in which the results were reported as numerical values. After the selection and data extraction process, the final list of studies was compiled through consensus.

### 2.4. Quality Assessment

Quality assessment of the enrolled studies was carried out using the Newcastle–Ottawa Scale (NOS) according to [57]. The quality assessment was based on the modified criteria of [58]. The possible scores ranged from 1 to 7, and studies that scored 7 were considered to be of the highest quality, with the lowest risk of bias. Studies that scored less than 7 were considered to be of lower quality, with a higher risk of bias.

We implemented a quality assessment procedure to take account of defects in various parameters. The deficiencies, such as a small sample size, the country of participants’ origin, absence of genderi nformation, age data, and method of analysis, were each scored as one “pointless”. ICP-MS was deemed the most representative analytical methodology, with all others receiving a score of one “unusable”. Finally, the number of points assigned to each study for each clinical matrix was determined, and a mean value for the entire meta-analysis was calculated. A detailed description of the quality assessment procedure can be found.

### 2.5. Statistical Analysis

The heterogeneity of the selected studies was evaluated using the I-squared (I^2^) and the associated Cochran’s Q test [59]. An I^2^ value exceeding 75% was considered indicative of a high level of heterogeneity, and to account for the limited power of the Q test in detecting heterogeneity, a significance level of *p* < 0.1 was used. In cases where heterogeneity exceeded 70%, pooled estimates were analyzed using the random effects model. We opted for the random effects model for all analyses, as we anticipated that the true effect sizes would vary among studies. Additionally, τ-squared (τ^2^), as per [60], was used to assess heterogeneity. A τ^2^ value close to 0 suggested low heterogeneity, while a τ^2^ value greater than 1 indicated substantial heterogeneity [61].

Effect sizes were calculated as mean differences in Hg levels in clinical materials (hair, whole blood, plasma, RBCs, and urine), and then converted to Hedges’s g, with adjustments to account for the influence of small sample sizes [62]. We also calculated 95% confidence intervals (CIs) to measure statistical variance in the pooled effect sizes. In addition, we determined the relative weight of each study to gain insight into the contribution of each study to the overall results of our meta-analysis. This was particularly important for studies categorized as outliers or those with a high risk of bias. The standard residual was also estimated to illustrate the unaccounted-for residual variability between studies. Significance was set at a two-sided *p* value of less than 0.05.

Statistical analysis was performed using Comprehensive Meta-Analysis software (v. 3.0, Biostat Inc., Frederick, MD, USA). Additionally, as previously mentioned, we employed VOSviewer 1.6.19 software to create network data maps for visualization and exploration.

### 2.6. Publication Bias

Publication bias is the selective publication of research studies, where studies with positive results are more likely to be published than studies with negative results. To avoid publication bias in the selection of publications for this meta-analysis, we included all studies that met the specified criteria, regardless of whether their outcome was positive or negative. We also performed appropriate statistical tests to mathematically calculate the publication bias. Although combining the data from independent studies using meta-analytical methods can improve statistical precision, it cannot altogether prevent bias.

To assess publication bias, we conducted Egger’s regression test [63] and Begg and Mazumdar’s rank correlation test [64]. For each type of clinical matrix, publication bias was visually represented using funnel plots. We also utilized the fail-safe method to determine the number of missing studies required to potentially improve the quality of the meta-analysis. However, these results are not presented in this context, given the fact that we did not find a statistically significant publication bias in any of the materials we examined.

## 3. Results

### 3.1. Study Selection and Identification

The process of study selection and identification is summarized in Figure 2. Our initial literature search across two primary databases, SCOPUS and PubMed, yielded a total of 9091 records. After removing 596 duplicate records, we further refined the selection based on title and abstract, resulting in the exclusion of 6525 records with irrelevant topics. The remaining 1374 reports were subjected to a detailed search for retrieval. Of these, 1129 reports were not retrievable, and the remaining 245 reports underwent eligibility screening. Within these 245 reports, 122 lacked the necessary data, 16 reports lacked control data, 43 reports were reviews, and 2 reports contained extremely abnormal data, all of which were excluded (total excluded, n = 183 reports). This process led to a final selection of 62 studies for inclusion in the analysis. After a final check, two additional studies with abnormal data were also excluded. Consequently, the meta-analysis included a total of 60 studies. The total number of control participants across all studies was 2991, while the total number of cases was 3892. This resulted in a cumulative total of 6883 participants considered for the meta-analysis.

### 3.2. Study Characteristics

Table 1 provides an overview of the characteristics of the studies incorporated into the meta-analysis. For the analysis of Hg level in hair, a total of 25 studies were included [65,66,67,68,69,70,71,72,73,74,75,76,77,78,79,80,81,82,83,84,85,86,87,88,89], encompassinf various geographic regions. Specifically, there were 7 studies conducted in Europe [70,71,74,76,80,85,88], 5 in North America [73,79,83,86,87], 9 in Asia [65,66,67,72,77,78,81,82,84], and 4 in Africa [68,69,75,89]. The meta-analysis of Hg levels in whole blood comprised 15 studies [78,86,90,91,92,93,94,95,96,97,98,99,100,101,102,103], with 2 conducted in Europe [85,92], 5 in North America [91,95,96,98,99], 1 in Central America [97], 5 in Asia [77,91,93,100,101], and 2 in Africa [94,102]. For plasma Hg levels, 6 studies were incorporated into the meta-analysis [62,92,103,104,105,106], with 2 originating from Europe [92,106], 2 from Asia [62,105], and 2 from Africa [103,104]. In the case of RBCs, the meta-analysis consisted of 5 studies [37,90,107,108,109], of which 2 were from North America [90,108] and 3 from Asia [37,107,109]. Finally, the analysis of urine Hg values included 9 studies [67,76,78,85,90,110,111,112,113], with 3 from Europe [76,85,111], 3 from North America [90,112,113], 2 from Asia [67,78], and 1 from Africa [110]. These studies collectively contributed to our analysis of Hg levels in various clinical matrices and were drawn from diverse geographic regions worldwide.

### 3.3. Quality Assessment

The quality scores assigned to the studies enrolled in the meta-analysis ranged from 1 to 7, with an average score of 6.17. The quality scores differed slightly across the different clinical matrices. Specifically, the quality score for hair studies averaged 6.20, for whole blood studies it was 6.47, for plasma studies it was 6.00, for RBCs studies it was 5.60, and for urine studies it was 6.56 (Table 2). Scores of 5 and 6 were assigned to specific studies, and they generally reflected certain criteria. A score of 5 was typically assigned when participant gender and age were not represented, when numerical data (such as standard deviation or error) were missing, and when ICP-MS was not used as the analytical method. A score of 6 was assigned to studies that did not report either participant gender or age but provided all necessary numerical data required for the meta-analysis. These scores did not imply that the selected studies were of lower quality but rather indicated that they did not fully meet the criteria established for this meta-analysis.

### 3.4. Meta-Analysis of Hg Levels in Hair

This portion of the analysis included 25 studies with a combined sample size of 1134 controls and 1361 cases. Among these studies, four reported age ranges for controls and cases, one study did not specify age, and the remaining 20 studies had mean ages of 5.64 years for controls and 5.72 years for cases. While four studies did not report the gender of participants, two studies only reported gender for cases, and the other 19 studies provided gender data for both controls and cases. Among these 19 studies, there were 481 girls and 623 boys in the control group, and 222 girls and 626 boys in the case group. Analytical techniques for Hg concentration assessment varied, with 16 studies using ICP-MS, eight studies using AAS, and one study using AFS.

The mean hair Hg levels showed considerable variation, ranging from 0.0077 ± 0.0039 µg/g [71] to 13.00 ± 12.68 µg/g [76] for controls, and from 0.127 ± 0.049 µg/g [70] to 8.26 ± 10.57 µg/g [76] for ASD cases. Out of the 25 studies, nine reported significantly higher Hg levels in the hair of cases compared to controls, eight reported significantly lower levels in cases, and eight reported no significant differences between the two groups.

Pooling of the data using the random effects model revealed no significant differences between cases and controls, with Hedges’s g = −0.432 (95% CI: −0.980, 0.115) and *p* = 0.122. Individual study effect sizes ranged from −47.909 (95% CI: −54.806, −41.012, *p* = 0.000) in the study by [88] to 1.548 (95% CI: 1.150, 1.945, *p* = 0.000) in the study by [83]. Relative weights and standard residuals for each study are presented in Figure 3. Relative weights ranged from 0.55% [88] to 4.36% [69], and standard residuals ranged from −12.65 [88] to 8.85 [89]. High heterogeneity was observed with I^2^ = 97.170%, Q(24) = 847.959, and τ^2^ = 1.772, *p* = 0.000, indicating substantial variation in the true mean effects between studies.

Publication bias was assessed using funnel plots, which indicated no significant publication bias. Egger’s regression test showed t_25_ = 1.027, *p* = 0.157, and Begg and Mazumdar rank correlation demonstrated Kendall’s τ = −0.120, *p* = 0.200 (Figure 4).

In summary, we did not provide evidence of higher Hg levels in the hair of ASD children compared to neurotypical children (*p* = 0.122).

### 3.5. Meta-Analysis of Hg Levels in Whole Blood

The meta-analysis of Hg levels in whole blood included 15 studies with a total sample size of 1019 neurotypical children and 1345 ASD children (Table 1). In two studies, age ranges were reported for both controls and cases; in one study, ages ranged from 2 to 5 years in both groups, and in another study, they ranged from 2.3 to 4.7 years for controls and 2.6 to 4.0 years for cases. For the remaining 11 studies, the mean age was 7.15 years for controls and 7.40 years for cases.

One study did not provide information about gender, while the other 14 studies reported that the control group consisted of 219 girls and 806 boys, and the case group included 315 girls and 896 boys. The analytical techniques used varied, with nine studies using ICP-MS and six studies using AAS (Table 1).

The mean Hg levels in whole blood varied markedly. In controls, they ranged from 0.00 ± 0.00 µg/L [94] to 19.53 ± 5.65 µg/L [77]. In cases, the levels ranged from 0.19 ± 0.62 µg/L [100] to 55.59 ± 52.56 µg/L [91]. Seven studies reported significantly higher Hg levels in the whole blood of cases than in controls, three reported levels significantly lower than in controls, while five studies did not find significant differences between the two groups.

The forest plot of pooled data under the random effects model (Figure 5) showed significant differences between cases and controls, with Hedges’s g = −0.813 (95% CI: −1.307, −0.318) and *p* = 0.001. Effect sizes in individual studies ranged from −10.438 (95% CI: −12.017, −8.859, *p* = 0.000) in the study by [94] to 0.532 (95% CI: 0.251, 0.813, *p* = 0.000) in the study by [100]. The relative weights and standard residuals for each study are also displayed in Figure 5. Relative weights ranged from 4.14% [94] to 7.07% [96], and standard residuals ranged from −7.92 [94] to 1.46 [100]. High heterogeneity was observed, with I^2^ = 96.654%, Q(14) = 418.462, and τ^2^ = 0.891, indicating substantial variation in the true mean effects between studies.

Publication bias was assessed using funnel plots, which indicated no significant publication bias. Egger’s regression test showed t_15_ = 1.621, *p* = 0.064, and Begg and Mazumdar rank correlation demonstrated Kendall’s τ = −0.276, *p* = 0.076 (Figure 6).

In conclusion, the pooled effect size indicates significantly higher Hg levels in whole blood among ASD cases compared to controls (*p* = 0.001).

### 3.6. Meta-Analysis of Hg Levels in Plasma

The meta-analysis of plasma Hg levels in controls and cases involved six studies with a combined sample size of 224 neurotypical children and 263 ASD children (Table 1). In one study, the mean age was not reported for females and was 2–6 years for males. In the remaining five studies, the mean age was 5.63 years for controls and 5.19 years for cases. The control group consisted of 83 girls and 141 boys, while the case group included 55 girls and 208 boys. Analytical techniques varied across studies, with one study using ICP-MS, two studies using ICP-OES, and three studies using AAS.

The mean Hg levels in plasma ranged from 0.00 ± 0.00 µg/L [106] to 12.08 ± 4.05 µg/L [104] in the control group, and from 0.81 ± 0.22 µg/L [105] to 32.90 ± 16.40 µg/L [104] in the case group. Three studies reported significantly higher plasma Hg levels in cases than in controls, one study found significantly lower levels, and two studies reported no significant differences.

The forest plot of pooled data under the random-effects model is depicted in Figure 7. The results show significant differences between the two groups, with Hedges’s g = −1.161 (95% CI: −2.247, −0.075) and *p* = 0.036. Effect sizes in individual studies ranged from −2.878 (95% CI: −3.535, −2.222, *p* < 0.001) in the study by [62] to −0.090 (95% CI: −0.402, 2.222, *p* = 0.571) in the study by [103], indicating high heterogeneity in plasma. The relative weights and standard residuals for each study are also shown in Figure 7. Relative weights ranged from 16.28% [106] to 17.19% [103], and standard residuals ranged from −1.37 [62] to 1.51 [104]. The heterogeneity was I^2^ = 96.256%, Q(5) = 133.562, and τ^2^ = 1.761, *p* = 0.000, indicating high heterogeneity in reporting Hg levels in plasma.

Funnel plots (Figure 8) were used to assess publication bias, which revealed no significant publication bias. Egger’s regression test showed t_5_ = 1.655, *p* = 0.087, and Begg and Mazumdar rank correlation demonstrated Kendall’s τ = −0.467, *p* = 0.094.

In summary, the pooled effect size indicates significantly higher plasma Hg levels in cases compared to controls (*p* = 0.036).

### 3.7. Meta-Analysis of Hg Levels in RBCs

The meta-analysis of Hg levels in RBCs involved five studies with a total sample size of 215 neurotypical children and 293 ASD children (Table 1). The mean age in the studies was 8.76 years for controls and 7.74 years for cases. Two studies did not report the gender distribution [37,109], and one study [107] reported the control group’s gender, but had an all-boys ASD group. The remaining control group consisted of 24 girls and 119 boys, while the case group included 11 girls and 207 boys. Analytical techniques used included ICP-MS in one study, AAS in three studies, and an Hg vaporimeter in one study.

Mean Hg levels in RBCs varied from 1.30 ± 0.20 µg/L [90] to 10.70 ± 4.30 µg/L [108] in the control group and from 1.20 ± 0.81 µg/L [90] to 22.20 ± 12.10 µg/L [108] in the case group. Four studies reported significantly higher Hg levels in the RBCs of cases compared to controls, while one study did not find significant differences.

The forest plot of pooled data under the random-effects model (Figure 9) indicated significant differences between cases and controls, with Hedges’s g = −1.354 (95% CI: −2.197, −0.512) and *p* = 0.002. Effect sizes in individual studies ranged from −2.736 (95% CI: −3.589, −1.882, *p* = 0.000) in the study by [37] to −0.099 (95% CI: −0.295, 0.493, *p* = 0.622) in the study by [90], showing high heterogeneity among RBCs. The relative weights and standard residuals for each study are also displayed in Figure 9. Relative weights ranged from 20.74% [90] to 21.03% [108]. The heterogeneity was high, with I^2^ = 93.974%, Q(4) = 66.383, and τ^2^ = 0.851, *p* = 0.000.

Publication bias was assessed using funnel plots, which revealed no significant publication bias. Egger’s regression test showed t_5_ = 0.901, *p* = 0.217, and Begg and Mazumdar rank correlation demonstrated Kendall’s τ = −0.200, *p* = 0.312 (Figure 10).

In conclusion, the pooled effect size indicates significantly higher levels of Hg in RBCs in cases compared to controls (*p* = 0.002).

### 3.8. Meta-Analysis of Hg Levels in Urine

The meta-analysis of urine Hg levels included nine studies with a total sample size of 399 neurotypical children and 623 ASD children (Table 1). In one study [78], the mean age ranged from 3 to 11 years for both groups, while in another study [76], the age was not specified. In the other seven studies, the mean age was 8.50 years for the controls and 7.53 years for the cases. The control group consisted of 76 girls and 233 boys, and the case group included 95 girls and 474 boys. The [112] study used only boys in their experiment, and the gender structure was not specified in the study by [76]. In eight studies, ICP-MS was used, and one study used AAS as the analytical technique.

Mean Hg levels in urine ranged from 0.29 ± 0.53 µg/g creatinine [112] to 5.40 ± 5.07 µg/g creatinine [111] in the control group and from 0.36 ± 0.62 µg/g creatinine [112] to 11.30 ± 6.63 µg/g creatinine [110] in the case group. Four studies reported significantly higher urine Hg levels in cases compared to controls, one study reported significantly lower levels, and four studies did not find significant differences.

The forest plot of pooled data under the random-effects model is depicted in Figure 11. The results showed significant differences between the two groups, with Hedges’s g = −0.471 (95% CI: −0.981, 0.040) and *p* = 0.071. Effect sizes in individual studies ranged from −0.333 (95% CI: −0.813, 0.147, *p* = 0.173) in the study by [113] to −2.092 (95% CI: −2.521, −1.663, *p* = 0.000) in the study by [110]. The relative weights and standard residuals for each study are shown in Figure 11. Relative weights ranged from 10.26% [85] to 12.61% [78]. Heterogeneity was present, with I^2^ = 92.204%, Q(8) = 102.612, and τ^2^ = 0.557, indicating a high degree of variation among studies. Compared with hair, whole blood, plasma, and RBCs, the least heterogeneity was obtained for studies reporting Hg levels in urine.

Funnel plots were used to assess publication bias (Figure 12), and they indicated no significant publication bias in urine samples. Egger’s regression test showed t_9_ = 0.246, *p* = 0.406, and Begg and Mazumdar rank correlation demonstrated Kendall’s τ = 0.000, *p* = 0.500.

In summary, the pooled effect size did not show significantly higher urine Hg levels in cases compared to controls (*p* = 0.071).

Overall, this meta-analysis revealed significantly higher Hg values in cases compared to controls in whole blood, plasma, and RBCs, with high heterogeneity observed. At the same time, no significant differences were found in hair and urine Hg levels between neurotypical children and ASD children. The analysis of publication bias for hair, blood, plasma, RBCs, and urine did not indicate statistically significant publication bias in any of the studies analyzed.

## 4. Discussion

In this section, we will not provide a detailed discussion of the studies enrolled in the meta-analysis, as presented in Table 1. Studies that provided specific data, such as Hg levels in clinical matrices of cases and controls sorted by gender, age, residence, and other demographic characteristics, will be discussed in more detail.

### 4.1. Hg in Hair

Scalp hair is considered a suitable sample for assessing Hg levels in ASD children. It is collected noninvasively, reflects long-term Hg exposure, and is a primary sample in evaluating the link between Hg and ASD [78,114]. However, it is important to collect and prepare hair samples appropriately to avoid potential contamination in the pre-analytical phase. Scalp hair grows at a rate of approximately 1.5 cm per month, and can provide insights into Hg exposure over time [115].

The 1999–2000 NHANES study reported a mean hair Hg value of 0.12 µg/g for healthy neurotypical USA children (n = 838, aged 1 to 5 years) [116]. Their study also highlighted that hair can contain significantly higher Hg levels compared to blood, making it a suitable sample for environmental science. When assessing hair from children with ASD, it is essential to consider the age of the children, as younger children with ASD could exhibit different Hg levels than older children with ASD. Majewska et al. [88] pointed out that younger children with ASD had lower Hg levels than older children with ASD. This discrepancy might be attributed to increased Hg exposure or variations in detoxification mechanisms over time. Nevertheless, it is crucial to explore these variables’ dynamics as children with ASD grow, depending on their Hg exposure and detoxification mechanisms.

Several studies reported significantly lower Hg levels in the hair of ASD cases compared to controls, and the specifics can be found in Table 1. A meta-analysis conducted by Saghazadeh and Rezaei [117], involving 1092 cases and 973 controls, did not establish a link between Hg and ASD in hair samples. However, it reported substantially higher hair Hg levels in ASD cases from developing countries compared to those from developed countries. These findings of unaltered levels of Hg in hair were consistent with the data we obtained in this meta-analysis of 1361 cases and 1134 controls. It is important to note that studies with exceptionally high Hg levels in hair, such as the one by Fido and Al-Saad [84], were excluded from our analysis.

To date, limited research has explored differences in hair Hg levels in ASD cases and neurotypical controls based on demographic and clinical factors. Adams et al. [73] found no significant differences in hair Hg levels for age groups 3–15 years and 3–6 years in the USA. In contrast, Zhai et al. [72] found significantly different hair Hg levels between female cases and female controls, but not between male cases and male controls in China. Geier et al. [53] reported a significant correlation between increasing ASD severity and higher hair Hg levels. However, research findings vary, as Lakshmi Priya and Geetha [82] did not find a significant association between ASD symptom severity and hair Hg levels. Fiore et al. [118] also reported no significant correlation between hair Hg levels and ASD symptom severity in participants from Catania, Southern Italy. Given these discrepancies, further extensive research is needed to better understand Hg levels in ASD children from different countries, accounting for demographic and clinical factors.

### 4.2. Hg in Whole Blood and Its Parts (Plasma/Serum and RBCs)

Whole blood and RBCs provide long-term information on Hg levels, particularly RBCs, which have a lifespan of approximately 120 days [31]. In contrast, plasma offers only short-term information on Hg exposure, making it less crucial as a clinical matrix in ASD [119]. Therefore, it is imperative to conduct comprehensive studies on the Hg status in the blood, hair, and urine of pregnant women, lactating women, and young children to gain in-depth insights into prenatal and early postnatal Hg exposure, potentially during the period when ASD begins to develop [41]. For instance, a study by Ryu et al. [120] found a connection between prenatal and early childhood Hg exposure and ASD behavior at age 5 by analyzing Hg levels from early pregnancy to age 3 in a longitudinal cohort study of 458 mother–child pairs.

Most studies have reported higher Hg values in whole blood, plasma, and RBCs of ASD cases compared to matched controls, particularly in children older than 3 years. For instance, Baj et al. [48] indicated higher serum Hg levels in 78.3% of children with ASD. Geier et al. [108] reported a twofold increase in Hg levels in RBCs of cases compared to gender- and age-matched controls. Nevertheless, a few studies have reported the opposite pattern, where Hg levels were lower in cases compared to controls (Table 1).

In addition to quantifying total Hg and Hg-species in whole blood and its derivatives, it is crucial to assess essential trace elements, primarily zinc (Zn) and selenium (Se). Some studies have indicated a deficiency of these two elements in ASD cases with elevated Hg profiles. For instance, Babaknejad et al. [121] reported a deficiency of Zn in ASD cases compared to controls, while El-Ansary et al. [37] found protective effects of Se in ASD cases with high Hg levels in RBCs. It is interesting to note that an occupational study by Chen et al. [122] with participants from an Hg-contaminated region reported significantly higher serum and urine Hg levels in the exposed population. McDowell et al. [116] found higher Hg levels in older women, women who consumed more seafood, and women of different ethnicities.

While a previous meta-analysis by Jafari et al. [35] reported significantly lower Hg levels in hair for cases compared to controls, they also reported higher Hg levels in whole blood and RBCs for ASD cases compared to controls. Shiani et al. [123] reported significantly higher Hg levels in 952 cases compared to 650 controls in their meta-analysis, which included 13 studies. These results corroborate previous findings of higher Hg levels in ASD cases compared to controls, but the current meta-analysis enrolled a substantially larger number of participants and samples, including plasma samples.

### 4.3. Hg in Urine

Urine reflects the short-term status of Hg exposure, and is easily accessible without invasive procedures [31,114,124]. Urine Hg excretion is particularly important in ASD cases receiving chelation therapy, such as dimercaptosuccinic acid (DMSA), which has been reported to lead to increased urine Hg excretion [113,125].

To date, three meta-analyses of urine Hg levels between cases and controls have been conducted. Jafari et al. [35] included eight studies (491 cases and 417 controls) and reported no significant differences in urine Hg levels between cases and controls. They also found no significant difference in urine Hg levels between cases in the US and certain other continents. Saghazadeh and Rezaei [117] similarly reported no significant differences between ASD cases and controls in terms of urine Hg levels. In contrast, Shiani et al. [122] conducted a meta-analysis involving seven studies (466 cases and 325 controls), and reported that children with ASD had significantly lower urine Hg levels compared to controls. Our meta-analysis observed no variations in urine Hg levels between cases and controls, consistent with the findings of Jafari et al. [35] and Saghazadeh and Rezaei [117], but on a larger sample.

As with hair and blood, there is limited knowledge about urine Hg levels in cases and controls concerning age, gender, and other demographic factors. We identified only one study that reported significantly lower urine Hg levels in children aged 2 to 4 years (n = 16, median value of 0.77 µg/L) compared to age-matched controls (n = 16, median value of 0.97 µg/L) [105]. In the comparison of urine Hg levels between children aged 4 to 6 years, no statistically significant differences in Hg levels were observed. This scarcity of data highlights the need for additional research in this direction to better understand the variations in urine Hg levels among different demographic groups.

In addition to measuring Hg levels in urine, several studies have focused on quantifying porphyrins in urine (pre-coproporphyrin, coproporphyrin, etc.), which could serve as specific biomarkers for Hg profiles in the body [38]. Porphyrins are heterocyclic compounds required for heme formation, an essential component of hemoglobin [48]. Many studies have reported significant increases in urine porphyrin levels in ASD cases compared to controls [35,77,104,112,126,127,128,129]. For instance, Geier and Geier [125] noted that children with ASD had 83% higher levels of urine coproporphyrin. Additionally, Geier et al. [130] established a link between high urinary porphyrin levels and ASD severity, although one study did not support this relationship [131]. This area warrants further research to enhance our understanding of urinary Hg and porphyrin levels in relation to ASD. In our meta-analysis, we did not include studies of children with ASD who underwent chelation therapy and subsequently had urinary Hg levels measured, which is actually evidence of the presence of a toxic metal in the organs.

### 4.4. Are Hg Levels in the Teeth a Promising Link to ASD?

Compared to other biological materials, deciduous teeth are perhaps the most promising clinical matrices for examining the link between Hg and ASD. Unfortunately, only two studies have been published on this topic to date, making it impossible to conduct a meta-analysis [41,112]. For a comprehensive understanding of the role of Hg in ASD, it is important to analyze tooth enamel rather than the entire tooth or dentin, because deciduous teeth’s enamel begins to develop in utero, and concludes between 3 months and 1 year after birth. This provides insight into prenatal and early postnatal exposure, a critical period when ASD begins to develop [112]. Of note, the Hg levels recorded by Adams et al. [41] for neurotypical children corresponded to levels found in brain tissue from monkeys subjected to thimerosal, simulating the US childhood vaccination schedule, emphasizing the importance of deciduous teeth, particularly enamel, in understanding the role of Hg in ASD. Hg levels ranging from 260 to over 600 ng/g have been reported in Minamata disease [41]. Further research in this area is needed to explore the potential relationship between Hg exposure through deciduous teeth and ASD.

### 4.5. Advantages and Limitations of Study Design and Further Directions

Although systematic review can provide useful overviews of the current state of knowledge on a topic if they are conducted with rigorous and clear methods, meta-analysis has some limitations. Although strict criteria were used to appropriately include individuals and exclude papers with very high levels of Hg in clinical matrices, the current study also had some limitations. We were unable to separate participants by gender, age, or residence, as the dimensionality required for reliable meta-analysis was lost. We have generalized results despite differences in primary research, combined different types of studies, and the summary effect may ignore important differences between studies, including the temporal relationship between exposure and outcome. In addition, it is possible that older children with ASD show more mouth behavior than healthy controls, leading to increased Hg levels in their biological tissues. There are not enough studies for nails and teeth, disease severity, and geographical region for ASD, and the measurement of total Hg but not inorganic or organic forms separately.

One of the primary challenges in elucidating the etiological role of Hg in ASD is the lack of access to the most authoritative clinical matrices to either confirm or refute a causal relationship. This primarily pertains to the inability to collect brain tissue from children with ASD due to the impracticality of surgical or biopsy procedures. The situation is further complicated by the fact that the exact timing of ASD onset remains unknown; symptoms emerge at varying ages.

For all the positive and negative aspects of the meta-analysis, we did our best to strictly follow all the rules of the PRISMA protocol to generate an appropriate study design to avoid heterogeneity, bias, and subjectivity. We are aware that this is not completely possible, but we hope that with this study, we have contributed to a better understanding of the relationship between Hg and ASD.

To address these limitations and further advance our understanding of the potential link between Hg and ASD, future investigations should consider the following:

Comprehensive Hg Analysis: Future studies should focus on detailed Hg analysis, including speciation and quantification (MeHg, EtHg, etc.), across a substantial number of participants diagnosed with ASD. These studies should ensure that participants are rigorously matched by factors such as gender, age, residence, diet, socio-economic status, and other uniform characteristics with neurotypical children.

Non-Invasive Clinical Matrices: To overcome the challenges of collecting brain tissue, researchers should emphasize non-invasive clinical matrices, such as hair, urine, and deciduous teeth, or less invasive clinical matrices (whole blood).

## 5. Conclusions

The present study provides valuable insights into Hg exposure and its potential link to ASD. Patients with ASD had higher whole blood, RBCs, and plasma levels of Hg, while Hg levels in hair and urine were unchanged. The findings support the hypothesis about the role of Hg as an environmental factor in the etiology of ASD. In addition, the findings of this study suggest that ASD children could have impaired excretory mechanisms for removing Hg from their bodies. Our results suggest that we must consider alternative explanations, such as different environmental exposure and increased deposition of Hg in other body tissues, which could lead to decreased excretion. We stress the promising avenue of investigating Hg levels in offering timely insights into MeHg and other Hg species’ impact on ASD. Furthermore, we emphasize the urgency of international collaboration to curtail environmental Hg exposure, amenable to non-invasive collection, offering timely insights into MeHg and other Hg species’ impact on ASD, highlighting the vital role of human–environment interaction in shaping future generations’ health. Through ongoing research and exploration, we aspire to unveil the intricate connection, if any, between Hg exposure and ASD.

Finally, we strongly recommended future research studies to examine the level of Hg in other biological materials, such as nail and deciduous teeth enamel, amenable to non-invasive collection, offering timely insights into MeHg and other Hg species’ impact on ASD.

Additional research is needed to shed light on the reliable reduction of Hg levels in the bodies of children with ASD and, thereby, reduce or prevent harmful effects.

## Figures and Tables

**Figure 1 biomedicines-11-03344-f001:**
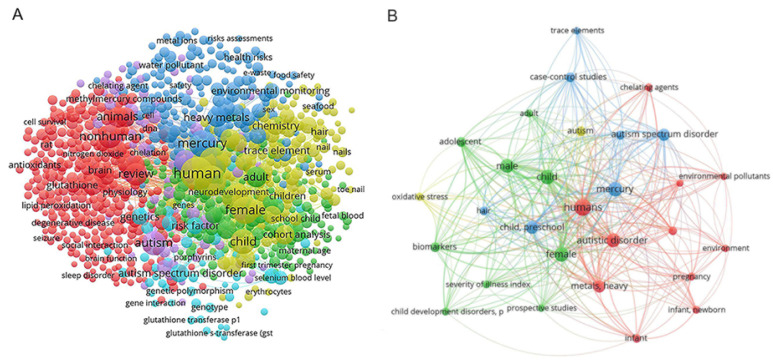
(**A**) Comprehensive network visualization of 1462 ASD-related keywords in scholarly literature; (**B**) focused network visualization of the 29 most commonly used ASD-related keywords in scholarly literature.

**Figure 2 biomedicines-11-03344-f002:**
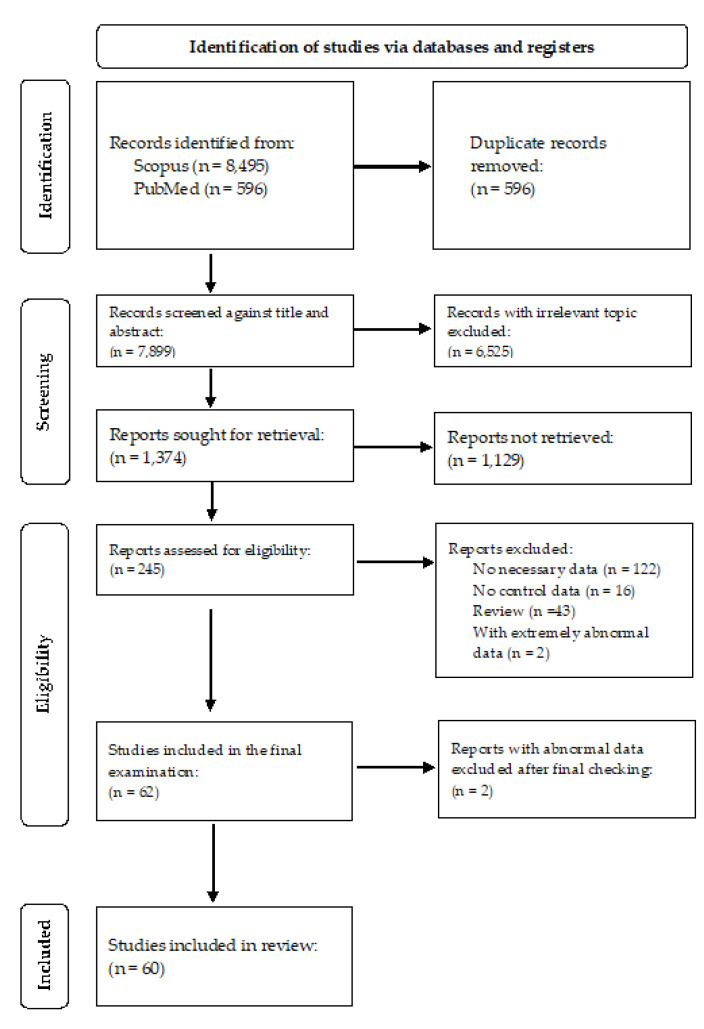
PRISMA flow diagram illustrating the literature search, study identification, inclusion, and exclusion Process. Abbreviation: n (number of studies).

**Figure 3 biomedicines-11-03344-f003:**
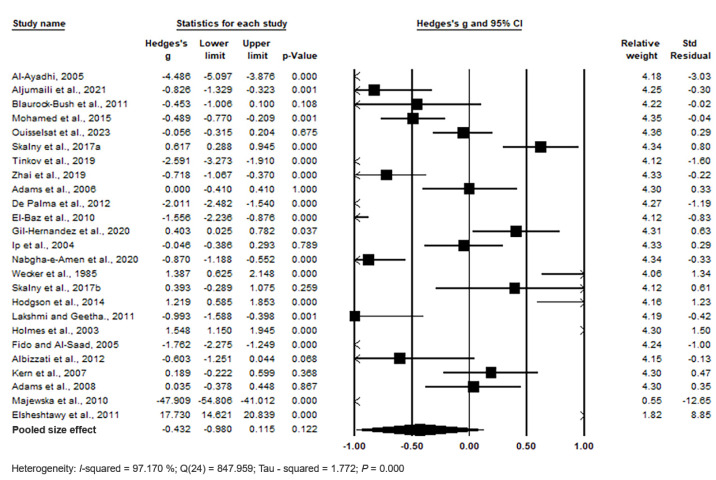
Forest plot for random-effects meta-analysis: variations in hair Hg levels between control children and cases. The size of each square corresponds to the study’s weight. The diamond symbol represents the overall pooled effect size for the studies included in the meta-analysis. Abbreviation: CI (Confidence Interval).

**Figure 4 biomedicines-11-03344-f004:**
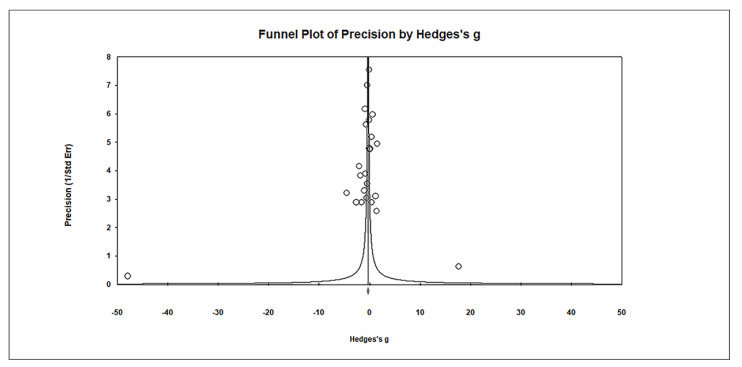
Funnel plots for publication bias assessment in studies comparing hair Hg levels in controls and cases. The plot displays the effect size (Hedges’s g) of the studies against their precision (inverse of SE). Observed studies are represented by circles, while the diamond symbol illustrates the overall pooled effect size based on these observed studies. Egger’s regression test: t_25_ = 1.027; *p* = 0.157; Begg and Mazumdar rank correlation: Kendall’s τ = −0.120, *p* = 0.200.

**Figure 5 biomedicines-11-03344-f005:**
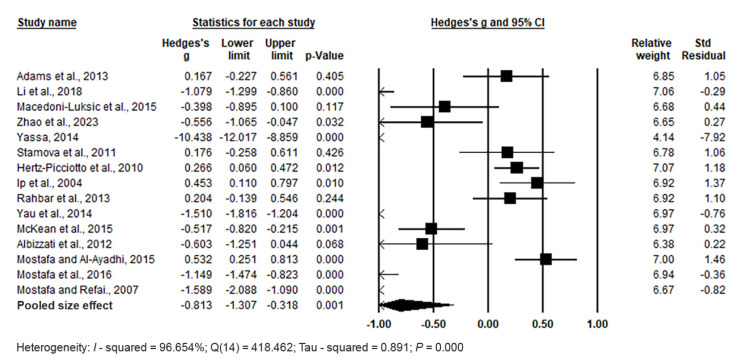
Forest plot for random-effects meta-analysis of differences in whole blood Hg levels between control children and cases. The size of each square corresponds to the study’s weight, with the diamond symbol representing the pooled total effect size for the studies included in the meta-analysis. Abbreviation: CI (Confidence Interval).

**Figure 6 biomedicines-11-03344-f006:**
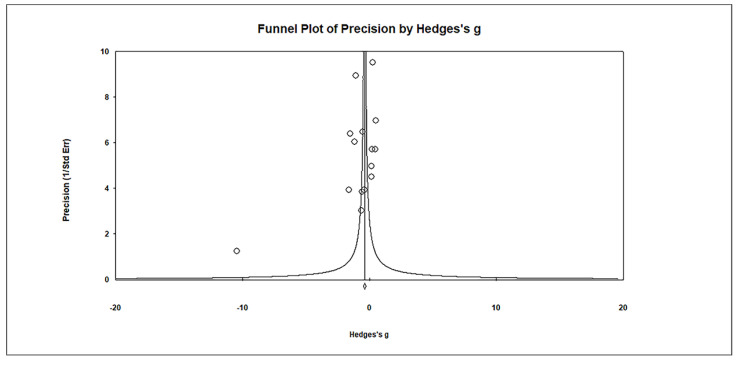
Funnel plots for assessing publication bias in observed studies comparing Hg levels in whole blood between control and case groups. The plot displays the effect size (Hedges’s g) of individual studies against their precision (inverse of SE). Observed studies are represented by circles, with the diamond symbol indicating the pooled overall effect size based on these observed studies. Egger’s regression test: t_15_ = 1.621, *p* = 0.064; Begg and Mazumdar rank correlation: Kendall’s τ = −0.276, *p* =0.076.

**Figure 7 biomedicines-11-03344-f007:**
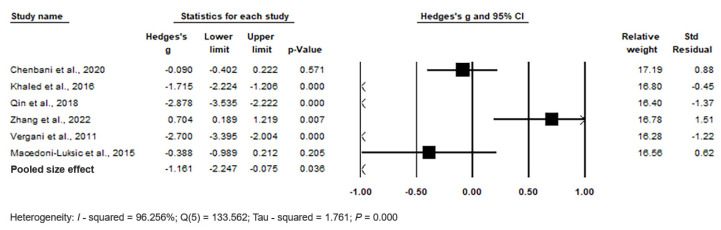
Forest plot for random-effects meta-analysis, depicting variations in plasma Hg levels between control children and cases. The size of each square corresponds to the study’s weight, and the diamond symbol represents the pooled total effect size based on the studies included in the meta-analysis. Abbreviation: CI (Confidence Interval).

**Figure 8 biomedicines-11-03344-f008:**
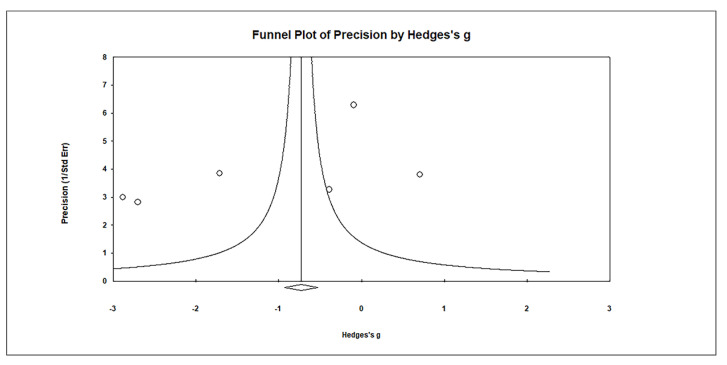
Funnel plots for evaluating publication bias in observed studies comparing Hg levels in the plasma of control and case Groups. The figure illustrates the effect size (Hedges’s g) of the studies against their precision (inverse of SE). Observed studies are represented by circles, and the diamond symbol represents the pooled overall effect size based on the observed studies. Egger’s regression test: t_6_ = 1.655, *p* = 0.087; Begg and Mazumdar rank correlation: Kendall’s τ = −0.467, *p* = 0.094.

**Figure 9 biomedicines-11-03344-f009:**
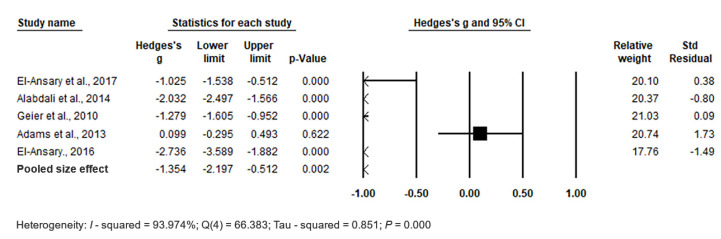
Forest plot for random-effects meta-analysis, depicting variations in Hg levels in red blood cells (RBCs) between control children and cases. Each square’s size corresponds to the study’s weight, and the diamond symbol represents the aggregated total effect size for the studies included in the meta-analysis. Abbreviation: CI (Confidence Interval).

**Figure 10 biomedicines-11-03344-f010:**
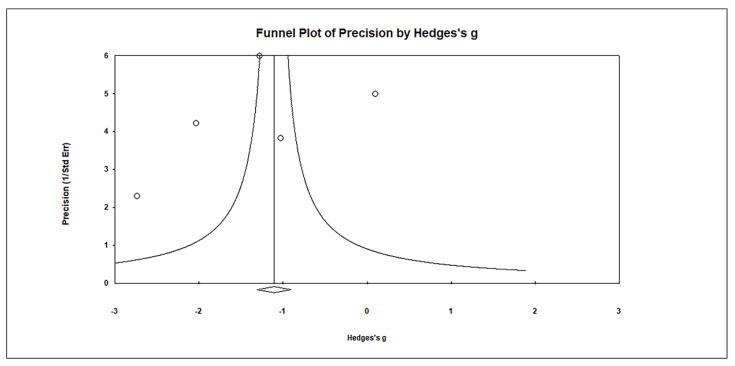
Funnel plots for assessing publication bias in observed studies comparing Hg levels in red blood cells (RBCs) of control and case groups. The figure illustrates the effect size (Hedges’s g) of the studies relative to their precision (inverse of SE). Circles represent individual observed studies, while the diamond symbol indicates the overall effect size derived from the observed studies. Egger’s regression test: t_5_ = 0.901, *p* = 0.217; Begg and Mazumdar rank correlation: Kendall’s τ = −0.200, *p* = 0.312.

**Figure 11 biomedicines-11-03344-f011:**
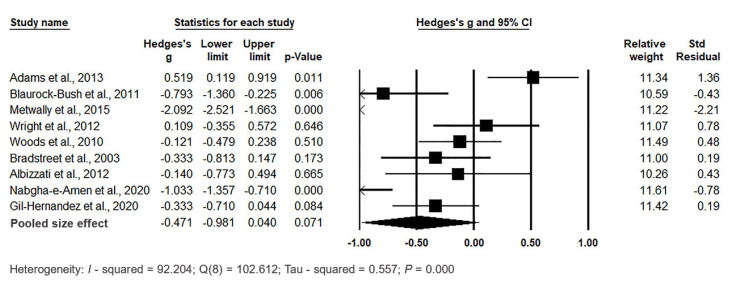
Forest plot for random-effects meta-analysis. Variations in urine Hg levels between control children and cases are presented. The size of each square corresponds to the study’s weight, while the diamond symbol represents the combined overall effect size for the studies included in the meta-analysis. Abbreviation: CI (Confidence Interval).

**Figure 12 biomedicines-11-03344-f012:**
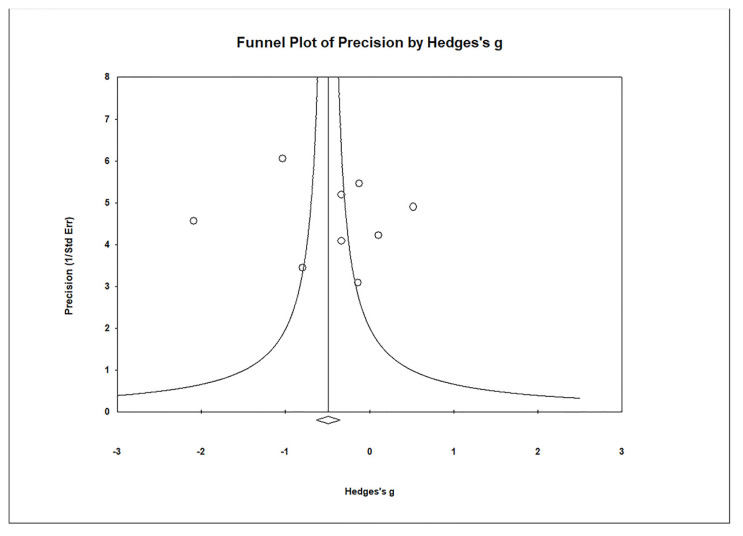
Funnel plots for the evaluation of publication bias in the examined studies, comparing Hg levels in the urine of control and case groups. The figure displays the effect size (Hedges’s g) of the individual studies in relation to their precision (inverse of the standard error). Each circle represents an observed study, and the diamond symbol represents the combined overall effect size calculated based on the observed studies. Egger’s regression test: t_9_ = 0.246, *p* = 0.406; Begg and Mazumdar rank correlation: Kendall’s τ = 0.000, *p* = 0.500.

**Table 1 biomedicines-11-03344-t001:** Overview of 60 studies (1985–2023) included in the comprehensive review and meta-analysis of Hg levels in controls and cases as follows: 25 studies (1985–2023) included in the systematic review and meta-analysis of Hg levels in hair (µg/g), 15 studies (2004–2023) included in the systematic review and meta-analysis of Hg levels in whole blood (µg/L), 6 studies (2011–2020) included in the systematic review and meta-analysis of Hg levels in plasma (µg/L), 5 studies (2010–2017) included in the comprehensive review and meta-analysis of Hg levels in red blood cells (RBCs) (µg/L), and 9 studies (2003–2020) included in the comprehensive review and meta-analysis of Hg levels in urine (µg/g creatine).

	Study	Country	Sample Size Controls/CaseS	Age Controls/Cases	Gender (Fem/Male) Controls/Cases	Biological Material	Analytical Technique	Mercury Level (Mean ± SD) Controls/Cases
Hair								µg/g
1.	Al-Ayadhi, 2005 [65]	Saudi Arabia	80/65	7.2 ± 0.7/9.0 ± 0.3	not specified; 4/61	Hair	AAS	0.713 ± 0.228/4.204 ± 1.129
2.	Aljumaili et al., 2021 [66]	Iraq	20/75	3–14/3–14	not specified	Hair	AAS	1.25 ± 0.66/3.44 ± 2.93
3.	Blaurock-Bush et al., 2011 [67]	Saudi Arabia	25/25	6.25 ± 2.31/5.29 ± 1.90	6/19; 3/22	Hair	ICP-MS	0.30 ± 0.31/0.47 ± 0.42
4.	Mohamed et al., 2015 [68]	Egypt	100/100	6.80 ± 3.04/6.20 ± 2.40	26/74; 16/84	Hair	AAS	0.25 ± 0.16/0.39 ± 0.37
5.	Ouisselsat et al., 2023 [69]	Morrocco	120/107	6.68 ± 2.39/7.14 ± 2.47	36/84; 25/82	Hair	ICP-MS	0.193 ± 0.13/0.200 ± 0.12
6.	Skalny et al., 2017 [70]	Russia	74/74	5.11 ± 2.34/5.12 ± 2.36	not specified	Hair	ICP-MS	0.167 ± 0.077/0.127 ± 0.049
7.	Tinkov et al., 2019 [71]	Russia	30/30	4.8 ± 2.2/4.7 ± 2.1	not specified	Hair	ICP-MS	0.077 ± 0.039/0.229 ± 0.072
8.	Zhai et al., 2019 [72]	China	58/78	4.90 ± 0.97/4.96 ± 1.01	27/31; 22/56	Hair	ICP-MS	0.26 ± 0.13/0.41 ± 0.25
9.	Adams et al., 2006 [73]	USA	40/51	7.5 ± 3.0/7.1 ± 3.0	10/30; 12/39	Hair	ICP-MS	0.29 ± 0.35/0.29 ± 0.41
10.	De Palma et al., 2012 [74]	Italy	61/44	8.4 ± 1.3/9.0 ± 4.0	25/36; 7/37	Hair	ICP-MS	0.25 ± 0.11/0.50 ± 0.14
11.	El-Baz et al., 2010 [75]	Egypt	15/32	5.53 ± 2.75/6.75 ± 3.26	6/9; 10/22	Hair	AAS	0.12 ± 0.019/0.79 ± 0.51
12.	Gil-Hernandez et al., 2020 [76]	Spain	54/54	Not specified	not specified	Hair	AAS	13.00 ± 12.68/8.26 ± 10.57
13.	Ip et al., 2004 [77]	Japan	55/82	7.8 ± 0.4/7.0 ± 0.2	9/46; 9/73	Hair	AAS	1.92 ± 1.58/1.98 ± 1.05
14.	Nabgha-e-Amen et al., 2020 [78]	Pakistan	76/90	3–11/3–11	22/54; 20/70	Hair	ICP-MS	1.0 ± 0.26/1.3 ± 0.4
15.	Wecker et al., 1985 [79]	USA	22/12	4.3 ± 2.6/5.67 ± 0.69	0/22; 0/12	Hair	AAS	15.75 ± 0.35/15.2 ± 0.45
16.	Skalny et al., 2017b [80]	Russia	16/16	5–8/5–8	0/16; 0/16	Hair	ICP-MS	0.151 ± 0.134/0.105 ± 0.09
17.	Hodgson et al., 2014 [81]	Oman	22/22	5.50 ± 1.00/3.5 ± 1.5	6/9; 7/15	Hair	ICP-MS	6.93 ± 0.36/6.03 ± 0.96
18.	Lakshmi and Geetha., 2011 [82]	India	15/50	4–12/4–12	not specified; 20/30	Hair	ICP-MS	0.37 ± 0.04/3.09 ± 0.37
19.	Holmes et al., 2003 [83]	USA	45/94	0.7 ± 0.102/0.7 ± 0.325	11/34; 21/73	Hair	ICP-MS	3.63 ± 3.56/0.47 ± 0.28
20.	Fido and Al-Saad, 2005 [84]	Kuwait	40/40	4.3 ± 2.67/4.2 ± 2.2	0/40; 0/40	Hair	ICP-MS	0.30 ± 0.24/4.50 ± 3.33
21.	Albizzati et al., 2012 [85]	Italy	20/17	10.41 ± 3.05/11.52 ± 3.20	6/14; 2/15	Hair	ICP-MS	0.28 ± 0.08/0.32 ± 0.04
22.	Kern et al., 2007 [86]	USA	45/45	3.0 ± 1.4/3.0 ± 1.4	10/35; 10/35	Hair	ICP-MS	0.16 ± 0.10/0.14 ± 0.11
23.	Adams et al., 2008 [87]	USA	31/78	1.37 ± 0.42/1.38 ± 0.37	11/21; 11/67	Hair	AFS	0.95 ± 0.87/0.87 ± 2.6
24.	Majewska et al., 2010 [88]	Poland	38/55	8.4 ± 0.20/8.1 ± 0.15	19/25; 19/30	Hair	AAS	0.14 ± 0.02/2.1 ± 0.05
25.	Elsheshtawy et al., 2011 [89]	Egypt	32/32	4.0 ± 0.8/4.1 ± 0.8	8/24; 8/24	Hair	ICP-MS	3.2 ± 0.2/0.55 ± 0.06
Whole Blood								µg/L
1.	Adams et al., 2013 [90]	USA	44/55	11.0 ± 3.1/10.0 ± 3.1	5/39; 6/49	Blood	ICP-MS	0.87 ± 0.76/0.75 ± 0.67
2.	Li et al., 2018 [91]	China	184/180	6.12 ± 1.69/5.06 ± 1.37	38/146; 30/150	Blood	AAS	13.47 ± 17.24/55.59 ± 52.56
3.	Macedoni-Lukšić et al., 2015 [92]	Slovenia	22/52	6.6. ± 3.7/6.2 ± 3.0	11/11; 6/46	Blood	AAS	1.55 ± 0.56/1.90 ± 0.97
4.	Zhao et al., 2023 [93]	China	30/30	4.2 ± 1.5/3.8 ± 1.3	15/15; 9/21	Blood	ICP-MS	0.685 ± 0.196/0.796 ± 0.198
5.	Yassa, 2014 [94]	Egypt	45/45	12.40 ± 2.04/11.30 ± 1.02	14/31; 13/32	Blood	ICP-MS	0.00 ± 0.00/4.02 ± 0.54
6.	Stamova et al., 2011 [95]	USA	51/33	2.3–4.7/2.6–4.0	0/51; 0/33	Blood	ICP-MS	0.6 ± 0.82/0.46 ± 0.73
7.	Hertz-Picciotto et al., 2010 [96]	USA	143/249	2–5/2–5	27/116; 28/221	Blood	ICP-MS	0.8 ± 1.3/0.49 ± 1.08
8.	Ip et al., 2004 [77]	Japan	55/82	7.8 ± 0.4/7.0 ± 0.2	9/46; 9/73	Blood	AAS	19.53 ± 5.65/17.68 ± 2.48
9.	Rahbar et al., 2013 [97]	Jamaica	65/65	2–8/2–8	not specified	Blood	ICP-MS	0.98 ± 0.79/0.83 ± 0.67
10.	Yau et al., 2014 [98]	USA, Mexico	78/149	6.6 ± 3.7/6.2 ± 3.0	16/133; 10/68	Blood	ICP-MS	0.32 ± 0.01/0.48 ± 0.13
11.	McKean et al., 2015 [99]	USA	58/164	2–8/2.8	22/36; 149/17	Blood	ICP-MS	4.29 ± 0.84/4.73 ± 0.85
12.	Albizzati et al., 2012 [85]	Italy	20/17	10.41 ± 3.05/11.52 ± 3.20	6/14; 2/15	Blood	ICP-MS	0.57 ± 0.34/0.67 ± 0.31
13.	Mostafa and Al-Ayadhi, 2015 [100]	Saudi Arabia	100/100	8.3 ± 1.6/8.1 ± 1.7	23/77; 22/78	Blood	AAS	0.43 ± 0.14/0.19 ± 0.62
14.	Mostafa et al., 2016 [101]	Saudi Arabia	84/84	7.0 ± 1.8/6.8 ± 1.5	24/60; 22/62	Blood	AAS	0.50 ± 0.14/0.8 ± 0.34
15.	Mostafa and Refai., 2007 [102]	Egypt	40/40	5.25 ± 1.80/5.38 ± 1.85	9/31; 9/31	Blood	AAS	3.9 ± 1.80/19.8 ± 13.9′
Plasma								µg/L
1.	Chehbani et al., 2020 [103]	Tunisia	70/89	7.81 ± 3.32/7.52 ± 3.02	29/41; 15/74	Plasma	AAS	0.77 ± 0.53/0.86 ± 1.24
2.	Khaled et al., 2016 [104]	Egypt	40/40	5.23 ± 1.25/4.12 ± 0.94	12/28; 8/32	Plasma	AAS	12.08 ± 4.5/32.9 ± 16.4
3.	Qin et al., 2018 [62]	China	38/34	4.29 ± 1.73/4.10 ± 0.81	17/21; 14/20	Plasma	ICP-OES	1.13 ± 1.05/3.89 ± 0.82
4.	Zhang et al., 2022 [105]	China	30/30	4.21 ± 0.93/4.03 ± 1.12	6/24; 6/24	Plasma	ICP-MS	0.96 ± 0.2/0.81 ± 0.22
5.	Vergani et al., 2011 [106]	Italy	32/28	Not specified/2–6	12/20; 7/21	Plasma	ICP-OES	0.00 ± 0.00/3.21 ± 1.72
6.	Macedoni-Lukšić et al., 2015 [92]	Slovenia	14/42	6.6 ± 3.7/6.2 ± 3.0	7/7; 5/37	Serum	AAS	1.55 ± 0.56/1.90 ± 0.97
RBCs								µg/L
1.	El-Ansary et al., 2017 [37]	Saudi Arabia	30/35	7.2 ± 2.14/7.0 ± 2.34	Not specified	RBCs	AAS	2.71 ± 0.57/3.66 ± 1.13
2.	Alabdali et al., 2014 [107]	Saudi Arabia	32/100	7.2 ± 2.0/7.0 ± 2.34	Not specified; 0/100	RBCs	AAS	5.12 ± 0.83/6.99 ± 0.94
3.	Geier et al., 2010 [108]	USA	89/83	11.4 ± 2.2/7.3 ± 3.7	19/70; 5/58	RBCs	Hg vaporimeter	10.7 ± 4.3/22.2 ± 12.1
4.	Adams et al., 2013 [90]	USA	44/55	11.0 ± 3.1/10.0 ± 3.1	5/39; 6/49	RBCs	ICP-MS	1.3 ± 1.2/1.2 ± 0.81
5.	El-Ansary., 2016 [109]	Saudi Arabia	20/20	7.4/7.4	Not specified	RBCs	AAS	4.64 ± 0.68/6.93 ± 0.94
Urine								µg/g creatinine
1.	Adams et al., 2013 [90]	USA	44/55	11.0 ± 3.1/10.0 ± 3.1	5/39; 6/49	Urine	ICP-MS	2.58 ± 1.10/1.01 ± 3.90
2.	Blaurock-Bush et al., 2011 [67]	Saudi Arabia	25/25	6.25 ± 2.31/5.29 ± 1.90	6/19; 3/22	Urine	ICP-MS	1.10 ± 0.63/2.48 ± 2.34
3.	Metwally et al., 2015 [110]	Egypt	75/55	4.02 ± 4.01	18/57; 16/39	Urine	ICP-MS	2.22 ± 0.35/11.3 ± 6.63
4.	Wright et al., 2012 [111]	UK	28/47	12.6 ± 3.5/9.6 ± 3.6	15/13; 10/37	Urine	ICP-MS	5.4 ± 5.07/4.97 ± 3.04
5.	Woods et al., 2010 [112]	USA	59/59	6.39 ± 3.06/6.01 ± 2.14	0/59; 0/59	Urine	ICP-MS	0.29 ± 0.53/0.36 ± 0.62
6.	Bradstreet et al., 2003 [113]	USA	18/221	8.85/6.25	4/14; 38/183	Urine	ICP-MS	1.29 ± 1.54/4.06 ± 8.59
7.	Albizzati et al., 2012 [85]	Italy	20/17	10.41 ± 3.05/11.52 ± 3.2	6/14; 2/15	Urine	ICP-MS	0.69 ± 0.07/0.70 ± 0.07
8.	Nabgha-e-Amen et al., 2020 [78]	Pakistan	76/90	3–11/3–11	22/54; 20/70	Urine	ICP-MS	1.0 ± 0.31/1.3 ± 0.27
9.	Gil-Hernandez et al., 2020 [76]	Spain	54/54	Not specified	Not specified	Urine	AAS	0.33 ± 0.42/0.54 ± 0.78

Abbreviations: ICP-MS (Inductively Coupled Plasma-Mass Spectrometry); ICP-OES (Inductively Coupled Plasma-Optical Emission Spectrometry); AAS (Atomic Absorption Spectrometry); AFS (Atomic Fluorescence Spectroscopy). Number of controls: 1134 (hair); 1019 (whole blood); 224 (plasma); 215 (RBCs); 399 (urine). Total number of controls: 2991. Number of cases: 1361 (hair); 1345 (whole blood); 263 (plasma); 293 (RBCs); 623 (urine). Total number of cases: 3892. Total number of participants enrolled in the meta-analysis: 6883.

**Table 2 biomedicines-11-03344-t002:** Quality assessment of included studies in the meta-analysis: Hg levels in ASD cases (based on the Newcastle–Ottawa Scale).

Study	Selection				Comparability	Outcome		Score	
	Representativeness	Size	Non Respondents	Exposure Determination	Design/ Analysis	Determination of Outcome	Statist. Test	For Sample Type Average	
Hair									
Al-Ayadhi, 2005 [65]	a	a	b	a	a	b	a	5	
Aljumaili et al., 2021 [66]	a	a	b	a	a	b	a	5	
Blaurock-Bush et al., 2011 [67]	a	a	a	a	a	b	a	6	
Mohamed et al., 2015 [68]	a	a	a	a	a	a	a	7	
Ouisselsat et al., 2023 [69]	a	a	a	a	a	a	a	7	
Skalny et al., 2017 [70]	a	a	b	a	a	a	a	6	
Tinkov et al., 2019 [71]	a	a	b	a	a	a	a	6	
Zhai et al., 2019 [72]	a	a	a	a	a	a	a	7	
Adams et al., 2006 [73]	a	a	a	a	a	a	a	7	
De Palma et al., 2012 [74]	a	a	a	a	a	a	a	7	
El-Baz et al., 2010 [75]	a	a	a	a	a	a	a	7	
Gil-Hernandez et al., 2020 [76]	a	a	c	a	a	b	a	4	
Ip et al., 2004 [77]	a	a	a	a	a	b	a	6	
Nabgha-e-Amen et al., 2020 [78]	a	a	a	a	a	b	a	6	
Wecker et al., 1985 [79]	a	a	a	a	a	b	a	6	
Skalny et al., 2017 [80]	a	a	c	a	a	a	a	5	
Hodgson et al., 2014 [81]	a	a	a	a	a	a	a	7	
Lakshmi and Geetha., 2011 [82]	a	a	c	a	a	a	a	5	
Holmes et al., 2003 [83]	a	a	a	a	a	a	a	7	
Fido and Al-Saad, 2005 [84]	a	a	b	a	a	a	a	6	
Albizzati et al., 2012 [85]	a	a	a	a	a	a	a	7	
Kern et al., 2007 [86]	a	a	a	a	a	a	a	7	
Adams et al., 2008 [87]	a	a	a	a	a	b	a	6	
Majewska et al., 2010 [88]	a	a	a	a	a	b	a	6	
Elsheshtawy et al., 2011 [89]	a	a	a	a	a	a	a	7	6.20
Whole blood									
Adams et al., 2013 [90]	a	a	a	a	a	a	a	7	
Li et al., 2018 [91]	a	a	a	a	a	b	a	6	
Macedoni-Lukšić et al., 2015 [92]	a	a	a	a	a	b	a	6	
Zhao et al., 2023 [93]	a	a	a	a	a	a	a	7	
Yassa, 2014 [94]	a	a	a	a	a	a	a	7	
Stamova et al., 2011 [95]	a	a	a	a	a	b	a	6	
Hertz-Picciotto et al., 2010 [96]	a	a	b	a	a	a	a	6	
Ip et al., 2004 [77]	a	a	a	a	a	a	a	7	
Rahbar et al., 2013 [97]	a	a	a	a	a	a	a	7	
Yau et al., 2014 [98]	a	a	b	a	a	a	a	6	
McKean et al., 2015 [99]	a	a	a	a	a	a	a	7	
Albizzati et al., 2012 [85]	a	a	a	a	a	a	a	7	
Mostafa and Al-Ayadhi, 2015 [100]	a	a	a	a	a	b	a	6	
Mostafa et al., 2016 [101]	a	a	a	a	a	b	a	6	
Mostafa and Refai., 2007 [102]	a	a	a	a	a	b	a	6	6.47
Plasma									
Chehbani et al., 2020 [103]	a	a	a	a	a	b	a	6	
Khaled et al., 2016 [104]	a	a	a	a	a	b	a	6	
Qin et al., 2018 [62]	a	a	a	a	a	b	a	6	
Zhang et al., 2022 [105]	a	a	a	a	a	a	a	7	
Vergani et al., 2011 [106]	a	a	b	a	a	a	a	6	
Macedoni-Lukšić et al., 2015 [92]	a	a	a	a	b	b	a	5	6.00
RBCs									
El-Ansary et al., 2017 [37]	a	a	b	a	a	b	a	5	
Alabdali et al., 2014 [107]	a	a	b	a	a	b	a	5	
Geier et al., 2010 [108]	a	a	a	a	a	a	a	7	
Adams et al., 2013 [90]	a	a	a	a	a	a	a	7	
El-Ansary., 2016 [109]	a	a	c	a	a	b	a	4	5.60
Urine									
Adams et al., 2013 [90]	a	a	a	a	a	a	a	7	
Blaurock-Bush et al., 2011 [67]	a	a	a	a	a	a	a	7	
Metwally et al., 2015 [110]	a	a	a	a	a	a	a	7	
Wright et al., 2012 [111]	a	a	a	a	a	a	a	7	
Woods et al., 2010 [112]	a	a	b	a	a	a	a	6	
Bradstreet et al., 2003 [113]	a	a	a	a	a	a	a	7	
Albizzati et al., 2012 [85]	a	a	a	a	a	a	a	7	
Nabgha-e-Amen et al., 2020 [78]	a	a	a	a	a	a	a	7	
Gil-Hernandez et al., 2020 [76]	a	a	c	a	a	b	a	4	6.56
6.17									

Selection: (1) Sample representativeness: a, truly representative of the average of the target population; b, reasonably representative of the average of the target population; c, selected group of users. (2) Sample size: a, satisfactory; b, unsatisfactory. (3) Nonrespondents: a, the comparability between the characteristics of the respondents and the nonrespondents is given, and the response rate is satisfactory; b, the response rate is not satisfactory, or the comparability between the respondents and the nonrespondents is not satisfactory; c, no description of response rate or the characteristics of the respondents and the nonrespondents. (4) Exposure determination: a, validated measurement instrument; b, measurement instrument not validated, but the instrument is available or described; c, no description of the measurement instrument. Comparability: (1) Comparability of subjects based on design or analysis: a, the study controls for the main factor; b, the study controls for each additional factor. Outcome: (1) Determination of outcome: a, independent blind assessment; b, record linkage; c, self-report. (2) Statistical test: a, the statistical test used to analyze the data is clearly described and appropriate, and the measure of association is stated, including confidence intervals and probability level (*p*-value); b, the statistical test is inappropriate, not described, or incomplete. Calculation: a = 1; b = from the maximum 7, 1 is subtracted; c = from the maximum 7, 2 is subtracted. Quality assessment was modified based on criteria as described by [58].

## Data Availability

The authors declare that all data supporting the findings of this study are available from the corresponding author upon reasonable request.

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
