# Peer review of "Mercury and Autism Spectrum Disorder: Exploring the Link through Comprehensive Review and Meta-Analysis"

_biomedicines, 2023, doi:10.3390/biomedicines11123344_

Round 1

Reviewer 1 Report

Comments and Suggestions for Authors

This comprehensive manuscript review and meta-analysis is perhaps one of the best I’ve seen. It is very well-written and carefully presented. There is not much more I can comment on. This work is an excellent overview of the role that Hg has in ASD. The authors have covered all aspects and implication of this association, from biological sequela to public health implications in a clear and eloquent way.

The methods are meticulously explained and thorough. I find figure 1 to be interesting, however, it is a bit blurry. Can this be fixed? The discussion is on point. It is difficult for me to see how this manuscript can be improved.

Author Response

We would like to thank the reviewer for taking the effort to read our manuscript and for his very kind words, which will encourage us in our further work. 

Point 1. „I find figure 1 to be interesting, however, it is a bit blurry. Can this be fixed“?

Response: Problems with Figure 1 A and B have been fixed (Page 5).

Reviewer 2 Report

Comments and Suggestions for Authors
General comments

This article reviewed and performed meta-analysis for 60 case-control studies to explore the link between mercury and autism. The article is well written in general and the methods are acceptable. However, authors do not explain why only select the case-control studies in the article. Some issues are not precise/accurate. Hg is classified as non-essential metal in Introduction and Discussion while in Search Strategy “heavy metals”, “toxic metal” are used. It is better to give the clear definition of non-essential metal, heavy metal, toxic metal and to use identical term throughout the entire text. Revision is necessary before further consideration.

 Specific comments

Introduction

Line 42: The reference [5] seems not directly relevant to the statement of “the incidence of ASD---- scientific literature”. Please find the original references.

Line 44-46: sentence “While the highest number---- less developed regions” is not relevant to the study aim and seems not logical given here.     

Line 56: give the definition and examples of non-essential trace metals.

Line 62: change “Hg” as “mercury (Hg)”.

Line 74: please check whether reference [31] is suitable for “ Further bioconversion--- the brain”

Line 77: for “ Hg easily cross the placental and blood-brain barriers”, please clearly indicate which form of Hg.

Line 102: I think thimerosal mainly contains ethylmercury?

Line 135-136: the aim is not clearly described. Which one potential aspect does this review article explore? Which target population. From the subsequent text, I understood that the target population is children, i.e. postnatal exposure? Please rewrite the aim.

Material and methods:

Line 158, please explain why the search the literature from 1985.

line 180: How “ sufficient numerical data” is defined?

Line 209-217: I think authors use Newcastle-Ottawa Scale (NOS) to do Quality Assessment. But in the text, NOS is not mentioned and NOS procedure needs more description, such as which score is classified as low-quality, etc.

Line 211: please explain how to define small sample size.

Results:

Table 1: last column is given as “ Lead level”, but the article focuses on Hg!!! Please check!

Discussion:

Line 581: please explain “case-only studies”. Are studies that only included ASD cases??

Line 628-629: please give the corresponding reference for “ In contrast, plasma offers only short-term information on Hg exposure”

Line 714-734: in the section of Limitations and further directions, authors mentioned causal relationship. However, the selected studies are case-control studies.  Is this design able to explore the causal relationship? I think the study design should also be discussed.

Author Response

First of all, we would like to thank the Editor and the Reviewers for their helpful suggestions during the review of this manuscript. Their suggestions have contributed notably to the quality of our manuscript. We have adopted all suggestions, changed parts of the text or provided appropriate explanations.

Thank you again for your effort to help us to improve our manuscript and your helpful comments.

Point 1. „...why only select the case-control studies...“

Response:  First, the case-control approach allows to observe a large group of people over a long period of time to collect enough cases. Secondly, the design of case-control studies allows for the simultaneous investigation of multiple risk factors. A case-control study provides a more practical approach. Case-control studies can also be very helpful when disease outbreaks occur and possible associations and exposures need to be identified. In a case-control study, the investigator can also include equal number of cases with controls. Because of these advantages, case-control studies are typically used as one of the first studies to demonstrate an association between an exposure and an event or disease.

Point 2. „It is better to give the clear definition of non-essential metal, heavy metal, toxic metal...“

Response: In the literature, the terms heavy metal and toxic metal are most commonly used for mercury. Using these terms as keywords will give us greater coverage of publications that refer specifically to the group of metals that include mercury than using the term non-essential metal. We emphasize that these are only terminological differences, that mercury is a non-essential and a heavy and a toxic metal, and that we have chosen the option that leads us to more search results.The following sentence is included in the text (Lines 82-83):  According to its role in the body, Hg is classified as a non-essential toxic element, while Hg is a heavy metal according to its physical and chemical properties. 

Point 3. „... and to use identical term throughout the entire text“.

Response:

We have accepted the reviewer's suggestion and will use the term “toxic” throughout the text. We have also define terms toxic- and heavy- metal (Lines 59-66, 82-83)

Point 4. Line 42. The reference [5] seems not directly relevant to the statement of “the incidence of ASD---- scientific literature”. Please find the original references.

Response: Reference 5 has been replaced as recommended by Reviewer 2 (Line 820). 

Point 5. Lines 44-46. sentence “While the highest number---- less developed regions” is not relevant to the study aim and seems not logical given here.

Response:

As recommended by reviewer this sentence has been removed.

Point 6. Line 56. give the definition and examples of non-essential trace metals.

Response:

We have given a definition and examples of non-essential trace elements (Lines 59-66).

Non-essential trace elements have no known function in the human body and can be toxic even in low concentrations. Non-essential trace elements include heavy metals and metalloids such as aluminum (Al), arsenic (As), cadmium (Cd), mercury (Hg), lead (Pb), antimony (Sb), tin (Sn), uranium (U), and vanadium (V). Their toxicity is related to their ability to damage vital organs such as the brain, kidney, liver and others. Long-term exposure to non-essential elements can lead to physical (e.g. chronic pain, changes in blood pressure, changes in blood composition, etc.) and psychological (e.g. anxiety, passivity, etc.) disorders, neurodegenerative diseases and cancer (Marquès et al., 2022)

Point 7 Line 62. change “Hg” as “mercury (Hg)”.

Response:  As recommended by Editor, introduction is condensed and this sentence is excluded from the text.

Point 8. Line 74. please check whether reference [31] is suitable for “ Further bioconversion--- the brain”

Response: Reference [31] is replaced with more appropriate. Now [30].

Point 9. Line 77. for “ Hg easily cross the placental and blood-brain barriers”, please clearly indicate which form of Hg.

Response:  We clearly indicate which forms of mercury. Thank you.

MeHg and EtHg are fat-soluble and have a high affinity for thiol groups, so that it can easily penetrate the placental and the blood-brain barrier.

Point 10. Line 102. I think thimerosal mainly contains ethylmercury?

Response: Corrected in the text (Lines 95-97).

Point 11. Line 135-136.... Please rewrite the aim.

Response: We rewrote the aim. Corrected in the text (Lines 168-173).

Material and methods:

Point 12. Line 158: please explain why the search the literature from 1985.

Response: In principle, all studies in our meta-analysis are between 2004 and 2023. Only one study (Wecker et al., 1985) relating to hair was included in our meta-analysis. As we defined certain criteria that the studies should fulfill, it was our wish to include as many studies as possible in the meta-analysis without overlapping with the meta-analyzes of other authors, or including some studies that other authors did not include.  

Point 13. Line 180. How “ sufficient numerical data” is defined?

Response: There are many studies that only report the mean value without the standard deviation or the standard error. There are also studies in which only graphs without numerical values are shown. „Sufficient numerical data“ refers to studies in which the mean ± standard deviation or another value from which the standard deviation can be calculated is precisely stated. 

Point 14. Line 209-217. I think authors use Newcastle-Ottawa Scale (NOS) to do Quality Assessment. But in the text, NOS is not mentioned and NOS procedure needs more description, such as which score is classified as low-quality, etc.

Response:  We explain and define criteria for quality assessment using Newcastle-Otawa scale in the text (Lines 244-248).

The quality assessment of the included studies was carried out using the Newcastle–Ottawa Scale (NOS) according to [Wells et al., 2009]. The quality assessment was based on the modified criteria of [Nakhaee et al., 2023]. The possible scores ranged from 1 to 7, and studies that scored 7 were considered to be of the highest quality and with the lowest risk of bias. Studies that scored less than 7 were considered to be of lower quality and with a higher risk of bias.

Point 15. Line 211. please explain how to define small sample size.

Response: In a statistical sense, we have accepted that a small sample is smaller if „n“ is less than 10 per group. In cases on studies in humans, there are very few such examples. We did not have such a case in our study, but we still wanted to distance ourselves from it in the text.

Results

Point 16. Table 1. last column is given as “ Lead level”, but the article focuses on Hg!!! Please check! Response:

Table 1. Corrected.

Discussion

Point 17. Line 581. please explain “case-only studies”. Are studies that only included ASD cases??

Response: Corrected: (Line 604). 

Point 18. Line 628-629. please give the corresponding reference for “ In contrast, plasma offers only short-term information on Hg exposure”

Response: We have added the appropriate reference (Lines 652, 1093).

Point 19. Lines 714-734. in the section of Limitations and further directions, authors mentioned causal relationship. However, the selected studies are case-control studies. Is this design able to explore the causal relationship? I think the study design should also be discussed

Response: As recommended by reviewer, we discuss study design and advantages and controversial aspects of meta-analysis and this part of text included in the sub-section Limitations and further direction (Lines: 736-738, 739, 742-748, 755-759)

Advantages and controversial aspects of study design Beside systematic reviews can provide useful overviews of the current state of knowledge on a topic if they are conducted with rigorous and clear methods, meta-analyzes have some limitations. The current study also has some limitations: generalizes results despite differences in primary research, combines different types of studies and the summary effect may ignore important differences between studies, the temporal relationship between exposure and outcome. In addition, it is possible that older children with ASD show more mouth behavior than healthy controls, leading to increased Hg level in their biological tissues, there are not enough studies for nails and teeth, differences in diagnostic criteria, disease severity and geographical region for ASD, measurement of total Hg but not inorganic or organic forms separately. For all the positive and negative aspects of the meta-analysis, we did our best to strictly follow all the rules of the PRISMA protocol to generate appropriate study design in order to avoid heterogeneity, bias and subjectivity. We are aware that this is not completely possible, but we hope that with this study we have contributed to a better understanding of the relationship between Hg and ASD. 

Reviewer 3 Report

Comments and Suggestions for Authors

The authors examined the role of mercury (Hg) in the autism spectrum disorder (ASD). They have reviewed 60 studies to assess the disparities in Hg levels betwee controls and ASD children. The studies examined Hg levels in five matrices: hair, whole blood, plasma,red blood cells (RBC) and urine. The authors show that Hg levels were significantly higer in ASD cases compared with controls when examined in whoole blood, plasma and RBC. No significant differences were detected in Hg levels in hair and urine between controls and ASD children.

I submit to the authors the following questions:
1) The elmination of toxic metals as Hg in the urine is provocked mainly by chelation therapy. In the absence of chelation, the deposition of Hg from blood into body tissues could lead to its decreased excretion. Indeed, only after removal of Hg from tissues by chelation therapy the toxic metals apppear in urine samples.

2) The presence of Hg in blood, plasma and RBC could indicate that the sources of Hg are still present and also that Hg has been not at all removed from blood into tissues.

3) The authors must specify that Hg is not the only  toxic metal resonsible for ASD.

4) What are for the authors the most important sources of Hg related to ASD, following the examination of the literature?

Comments on the Quality of English Language

Quality of English Language is good

Author Response

Responses to editors' and reviewers' comments

First of all, we would like to thank the Editor and the Reviewers for their helpful suggestions during the review of this manuscript. Their suggestions have contributed notably to the quality of our manuscript. We have adopted all suggestions, changed parts of the text or provided appropriate explanations.

Thank you again for your effort to help us to improve our manuscript and your helpful comments. 

Point 1.

Response: We agree with the reviewer and have stated in our conclusion that one of the reasons for the reduced excretion of mercury via the urine is its deposition in the organs. An important factor in the release of mercury from the organs is chelation therapy. Recent studies (Bridle et al., 2021) show that nearly 90% of the CH3Hg+ located in organ tissues of mice can be mobilized to urine by the oral administration of N-acetyl-L-cysteine (NAC). 

Point 2.

Response: We also agree with the statement of the reviewer. According to literature sources, mercury remains in the blood for a relatively short time, binds to a large number of low molecular mass proteins, deposited in organs, and crosses the blood brain barrier and the placenta. The presence of mercury in the blood indicates that the organism could be constantly exposed to sources of mercury, e.g. through food, environmental pollution, etc. 

Point 3.

Response: This statement is already included in the text (Lines.. - ..): „….there is a growing suspicion that non-essential trace metals, particularly those with neurotoxic properties, could play a crucial role in the etiology of ASD“ (Lines 57-58). 

Point 4.

Response: According to the literature, the main sources of Hg related to ASD are as follows:·        

Environmental sources: As a ubiquitous natural element, Hg passes from the air as vapor into water or soil, where it is either vaporized or converted to an organic form. Hg vapor enters the air from various natural and industrial sources, such as volcanic eruptions, gold refining and coal burning. Environmental Hg spreads far from the original source and eventually enters the biosphere.·    

The vaccine preservative thimerosal (ethylmercury).·        

Laboratory accidents and other accidental exposures, e.g. from the breakage of old thermometers.·        

Dental amalgam can be a significant source of inorganic Hg, which enters the bloodstream via inhalation of Hg vapor produced, for example, during chewing. 

All of these sources of mercury exposure can occur prenatally to mothers who absorb mercury into the body, which is then transferred to the fetus via the placenta, and some of them postnatally by exposing newborns and infants during the intensive phase of neurological development. 

Above mentioned statements are included in different parts of the text.

Round 2

Reviewer 2 Report

Comments and Suggestions for Authors

The manuscript is improved.  Minor revision is needed. For example, authors clarify the reason why only selecting case-control studies and explain “sufficient numerical data” in the author's reply. These explanations should be given in the manuscript.

Line 238: Abbreviation SEM  should be defined.    

Author Response

We would like to thank the reviewers once again for their helpful suggestions during the review of this manuscript. Their suggestions contributed significantly to the quality of our manuscript. We accepted all suggestions made by Reviewer 2 and Reviewer 3 in the second round of the reviewing process. Due to one of Reviewer 3's requests, we had to change the abstract to adapt it to the journal's propositions (200-250 words). Added text parts in R2 are green (R1 was in blue). The abstract was also changed with the Track changes option.

Responses to the Reviewer 2

Thank you for re-reading the revised text of the manuscript and making further helpful suggestions that will certainly improve the quality of the manuscript and make it easier for readers to understand. We have accepted all of the reviewer's suggestions.

Topic 1. “The manuscript is improved. Minor revision is needed. For example, authors clarify the reason why only selecting case-control studies and explain “sufficient numerical data” in the author's reply. These explanations should be given in the manuscript”.

Response. Thank you. The explanation is given in the text.

Studies that refer to sufficient numerical data include results where the mean ± standard deviation or standard error, or some other numerical value from which the standard deviation can be calculated, is accurately reported. In fact, there are many studies that report only the mean without the standard deviation or standard error, or studies in which only graphs without numerical values are shown. In our meta-analysis, we set the criterion of using only complete data, i.e. mean values and standard deviations that represent sufficient numerical data for us.

Topic 2. Line 238: Abbreviation SEM should be defined.

Response. Abbreviation SEM is defined in green color in parentheses. Thank you.

Reviewer 3 Report

Comments and Suggestions for Authors

The authors have to clarify these concepts:

1) They have to underline (in the abstract also) that the papers examined  lack of the very important contribution due to results obtained in ASD children subjected to chelation therapies, followed by measure of Hg levels in urine samples: this measure indicates really the presence of the toxic metal in the organs!!!

2) Thay have to explain that Hg is not the only one toxic metal related to ASD

Comments on the Quality of English Language

English language is good

Author Response

We would like to thank the reviewers once again for their helpful suggestions during the review of this manuscript. Their suggestions contributed significantly to the quality of our manuscript. We accepted all suggestions made by Reviewer 2 and Reviewer 3 in the second round of the reviewing process. Due to one of Reviewer 2's requests, we had to change the abstract to adapt it to the journal's propositions (200-250 words). Added text parts in R2 are green (R1 was in blue). The abstract was also changed with the Track changes option.

First of all, we would like to thank Reviewer 3 for his suggestions and ideas for improving this manuscript. We have accepted all of the reviewer's suggestions.

Topic 1. The authors have to clarify these concepts:

1) They have to underline (in the abstract also) that the papers examined  lack of the very important contribution due to results obtained in ASD children subjected to chelation therapies, followed by measure of Hg levels in urine samples: this measure indicates really the presence of the toxic metal in the organs!!!

Response. As recommended by the Reviewer, we have included additional explanations about the topic. In the abstract and in the text, we have emphasized that we have not included any studies in which chelation therapy was used in children with ASD. 

In Abstract. “This meta-analysis did not include the data of children with ASD who received chelation therapy“.

In section: 4.3. Hg in urine

In our meta-analysis, we did not include studies of children with ASD who underwent chelation therapy and who subsequently had urinary Hg levels measured, which is actually evidence of the presence of a toxic metal in the organs.

Topic 2.

2) Thay have to explain that Hg is not the only one toxic metal related to ASD.

Response. As suggested by the Reviewer 3, we have explained that Hg is not only one toxic metal related to ASD.

It is known that Hg is not the only toxic metal associated with ASD. Other metals related to ASD are aluminum (Al), antimony (Sb), arsenic (As), beryllium (Be), cadmium (Cd), chromium (Cr), lead (Pb), and nickel (Ni) ) [17,18].

Round 3

Reviewer 3 Report

Comments and Suggestions for Authors

The final version of the paper is suitable for publication